# High Sensitivity Snapshot Spectrometer Based on Deep Network Unmixing

**DOI:** 10.3390/s20247038

**Published:** 2020-12-09

**Authors:** Hui Xie, Zhuang Zhao, Jing Han, Lianfa Bai, Yi Zhang

**Affiliations:** School of Electronic Engineering and Optoelectronic Technology, Nanjing University of Science and Technology, Nanjing 210094, China; xiehui@njust.edu.cn (H.X.); eohj@njust.edu.cn (J.H.); blf@njust.edu.cn (L.B.); eo_zhy441@njust.edu.cn (Y.Z.)

**Keywords:** Hadamard transform spectrometer, snapshot HTS, neural network

## Abstract

Spectral detection provides rich spectral–temporal information with wide applications. In our previous work, we proposed a dual-path sub-Hadamard-s snapshot Hadamard transform spectrometer (Sub-s HTS). In order to reduce the complexity of the system and improve its performance, we present a convolution neural network-based method to recover the light intensity distribution from the overlapped dispersive spectra, rather than adding an extra light path to capture it directly. In this paper, we construct a network-based single-path snapshot Hadamard transform spectrometer (net-based HTS). First, we designed a light intensity recovery neural network (LIRNet) with an unmixing module (UM) and an enhanced module (EM) to recover the light intensity from the dispersive image. Then, we used the reconstructed light intensity as the original light intensity to recover high signal-to-noise ratio spectra successfully. Compared with Sub-s HTS, the net-based HTS has a more compact structure and high sensitivity. A large number of simulations and experimental results have demonstrated that the proposed net-based HTS can obtain a better-reconstructed signal-to-noise ratio spectrum than the Sub-s HTS because of its higher light throughput.

## 1. Introduction

Spectral detection is widely used in industry, remote sensing, military and many other aspects. However, the traditional slit spectrometer has a contradiction between fast acquisition and high signal-to-noise ratio (SNR). If we want to improve the response speed of the system, it will inevitably reduce the acquisition time of the system, resulting in a decrease in the system SNR. In order to improve the SNR of detected spectra, researchers have introduced compressed sensing (CS) [1,2,3], computational slits [4,5] and deconvolution [6].

Compressed sensing is a representative method to realize snapshot acquisition. Compressed sensing was first proposed by Donoho, Candes and Tao in 2006 [1,2]. In the same year, Brady et al. of Duke University proposed a new spectral imaging technology called coded aperture snapshot spectral imager (CASSI) and developed several CASSI-based snapshot imaging spectrometers in the following years [7,8]. In the aspect of reconstruction algorithms, a greedy algorithm [4,5,6], L1 norm convex optimization algorithm [9,10,11], L1 norm non-convex optimization algorithm [12], Bayesian method [13,14,15] and other reconstruction algorithms have been developed. However, these reconstruction algorithms all have some drawbacks. Although the greedy algorithm has low computational complexity, the reconstruction effect is not ideal. The L1 norm minimization method has good reconstruction performance, but it has high computational complexity. The Bayesian method lacks a strict theoretical guarantee. Thus, we should develop new reconstruction algorithms.

Based on the idea of compressed sensing, researchers proposed a static multimode multichannel spectrometer (MMS), which takes advantage of the noise reduction performance of Hadamard coding and uses the non-negative least squares method to directly reconstruct the spectrum [16,17,18]. This method can achieve snapshot measurement and maintain the advantage of high light throughput. However, compressed sensing is an ill-posed inverse problem, and the measurement results are not stable enough, which limits its application.

Compared to multiplexing methods, these numerical methods belong to so-called ill-posed inverse problems and cannot obtain certain and robust SNR improvement. In our previous work [19], we proposed a dual-path sub-Hadamard-s snapshot Hadamard transform spectrometer (Sub-s HTS) based on the Hadamard transform spectrometer (HTS) [20,21]. In this Sub-s HTS, all incident light is encoded simultaneously by the encoding matrix, and an extra imaging path is designed to measure the intensity distribution of the scene. The spectral measurement problem is transformed into a positive definite problem so that a stable and reliable SNR enhancement can be obtained and maintain snapshot.

However, there are some problems in our previous work [19]. Firstly, the simultaneous acquisition of light intensity and dispersion image requires a dual optical path system, which will reduce the overall light throughput by half and decrease the SNR. Secondly, dual-camera images require pixel-level registration which will cause a lot of problems and affect the reconstruction results. Finally, the dual optical path system increases the overall system size and reduces reliability. Therefore, how to use a single camera to complete the reconstruction process while ensuring the quality of the reconstructed spectrum becomes a problem that must be solved.

In recent years, the convolution neural network has shown strong fitting ability in many fields. In order to improve the light throughput of the system and reduce its complexity, we designed a convolution neural network to recover the light intensity distribution through the overlapped dispersive spectra and realize the single-path high-sensitivity snapshot spectral measurement.

The main contributions of this paper are as follows:(1)A light intensity recovery network (LIRNet) to solve the problem of spectral image unmixing and realize the direct acquisition of light intensity distribution through an overlapped dispersive spectral image;(2)The feasibility of using reconstructed light intensity data for sub-Hadamard matrix spectral detection is proven;(3)Both simulated and experimental results demonstrated that the performance of the net-based scheme can achieve similar or even better reconstructed results compared with the dual-path scheme and can improve the compact of the system.

## 2. Design of Snapshot Spectrum Detection Framework

In our previous work [19], we proposed a dual-path snapshot Hadamard transform spectrometer. The implementation and scheme of the snapshot spectrometer are shown in Figure 1a,b, respectively.

The snapshot HTS contains two imaging paths: non-dispersive and dispersive imaging paths. The non-dispersive imaging path is employed to capture the light intensity at the coding aperture. The dispersive imaging path is employed to capture the overlapped dispersive spectra. We take the 7 × 7 Hadamard matrix as an example to explain the spectral dispersion overlapping process in the snapshot spectrometer and show it in Figure 2.

In Figure 2, $S_{1}$ is the 1st row of the Hadamard-S matrix, $I_{1}$ is the 1st row light intensity distribution of the scene, $f_{1}$ is the spectrum of the 1st row, $f_{11}$ is the spectrum of the 1st pixel in the 1st row and m represents its range and the zeros in the spectrum represent shift invariant, which makes processing easy to understand. The measurement problems of the *i*th row of snapshot HTS are as follows:(1)$$g_{i}=\left(S_{i}\circ I_{i}\right)f_{i}+n_{snap_{i}}$$
where $g_{i}$ is the overlapping dispersion spectrum of the *i*th row, $S_{i}$ is the *i*th row of Hadamard-S matrix, $I_{i}$ is the *i*th row of light intensity distribution, $f_{i}$ is the *i*th row of the spectrum to be measured and $n_{snap_{i}}$ is the measurement noise in *i*th row measurement. If each row of the scene has the same spectrum, or we want to measure the average spectrum of each column, i.e., $f_{1}=f_{2}=\cdots =f_{n}$ or $f_{11}=f_{21}=\cdots =f_{n1}=\frac{1}{n}\sum_{i=1}^{n}f_{i1}$. Based on this assumption, the whole measurement of snapshot HTS can be simplified as:(2)$$g=\left(S\circ I\right)f+n_{snap}=S_{snap}f+n_{snap}=\left(S-S_{h}\right)f+n_{snap}$$
where $S_{snap}$ is the normalized modulation intensity distribution (sub-Hadamard-s matrix or Sub-s matrix) and $S_{h}$ is the difference between Hadamard-S matrix and normalized modulation intensity distribution. Thus, once the light intensity distribution $S_{snap}$ is obtained, the spectra need to measure can be recovered through an inverse process, otherwise, we cannot recover the spectra.

As shown in Figure 2, the extra non-dispersive imaging path is added to measure the light intensity distribution. The extra imaging path increases the complexity of the system and weakens the intensity of the measured overlapped spectra. If we can recover the light intensity from the measured overlapped spectra such as the red arrow in Figure 1b, we can remove the extra imaging path and improve the performance of the system. We designed a light intensity recovery network (LIRNet) to recover the light intensity from the dispersive image. The overall structure of the network is shown in Figure 3.

In LIRNet, we first designed an unmixing module (UM) to approximate the inverse process of convolution in the following Equation (3) which reduced the mixture of input pixels and output coarse unmixed spectral images. We then used an enhanced module (EM) to reconstruct the light intensity for sharper edges and better details.

## 3. Network Setup

As shown in Figure 3, in the raw spectral image, the intensity dispersion of each point is overlapped by grating, and the captured spectral image can be expressed as the overlapped image of each band spectral image. This process can be expressed by the convolution operator. Each pixel in the spectrum can be represented as:f = conv(S, k)(3)
where f is a single pixel in the spectral image, k is a one-dimensional convolution kernel in the spectral direction, the kernel size of convolution is the number of spectral bands and S is a high-dimensional spectral image.

From the perspective of spectral dimension, the light intensity of a point is the sum of its dispersive spectral intensity:(4)$$f_{s}=\overset{n}{\underset{i=1}{\sum }}S_{i}$$
where n is the number of spectral bands. This optical process is equal to the overlapping of a spectral image.

The UM contains five convolution layers and a deconvolution layer which extracts the primary features at the spectral dimension. The EM is an encoder–decoder network output feature map of UM, a further enhancement to supplement more visual information such as image contrast, brightness and texture features.

While training the network, we used two types of the loss function, intensity loss (IL) and spectrum loss (SL). IL adopts the mean square error (MSE) based on the online hard example mining (OHEM) [22] strategy as a loss function and SL adopts the SNR of the reconstructed spectrum as another loss function.

Euclidean distance is expressed as $D_{i}=||x_{i}-{\hat{x}}_{i}||{}^{2}$. The total number of samples is *N*, then:(5)$$Loss\left(x_{i},{\hat{x}}_{i}\right)=\frac{1}{N}\overset{N}{\underset{i=1}{\sum }}D_{i}$$
where $x_{i},{\hat{x}}_{i}$ are real and reconstructed samples respectively. In this equation, samples with higher (high-frequency areas) and lower (low-frequency areas) errors are averaged for network training, which makes it difficult for the optimizer to optimize in the training stage. To make the optimizer focus on the harder samples, Wu et al. put forward a method of hard-sample mining for pixel-level classification tasks [22]. We adopted the idea and modified it to adapt our intensity approximation task. We selected pixel samples with higher loss values as hard negative samples for training and ignored others with lower loss values so that the network would not repeat learn samples in low-frequency areas and would improve the performance in high-frequency areas. In this case, the loss function can be expressed as:(6)$$Loss=\frac{1}{\sum_{i}^{N}1\left\{D_{i}<t\right\}}\overset{N}{\underset{i}{\sum }}1\left\{D_{i}<t\right\}D_{i}$$
where *t* is the threshold, which is the minimum loss value of the first half of the sample with a larger loss value.

In order to make the intensity approximation consistent with realistic physical processes better, we used spectrum loss to further fine-tune the network:(7)$$Loss_{s}=R-SNR\left(recon\left(\hat{x}\right),f\right)$$
where R is the reversal factor, *recon* is the reconstruction processing of overlapped spectrum and $\hat{x}$, *f* is the real spectrum.

We used 1650 groups of multispectral images with 127 × 127 size for training, 350 groups for validation and 200 groups that were not involved in the training for testing. The images are padded to 128 × 128 to train conveniently. Finally, a 128 × (128 + *n*) overlapped spectral image was constructed where *n* is the number of spectral bands. The light intensity images that were superimposed by all bands were used as the training label.

In the UM, the convolution layer with 1 × *n* convolution kernel was used for unmixing, and the convolution layer with 3 × 3 convolution kernel and rectified linear unit (ReLU) [23] activation function was used to extract the primary features. In the enhancement module, the encoder–decoder network was used to extract the low-level and high-level semantic information of the unmixed feature map to enhance the unmixing image. The design of the enhance module structure is referred to as ERFNet [24], as shown in Table 1. The down-sampler layer is a pooling down-sampling process. The Non-bt-1d is the factorized convolution which decomposes the 3 × 3 convolution into a pair of 1D convolutions, which constitutes a non-bottleneck 1D structure as shown in Figure 4. With a 33% reduction of parameters, we can achieve the same learning ability and accuracy as the traditional non-bottleneck structure, improve the network efficiency and contribute to real-time spectral analysis.

## 4. SNR Analysis

In order to analyze the denoise performance of the net-based HTS, we set the sub-Hadamard matrix normalized intensity distribution as $S_{snap}$; $S_{1}$ is the difference between the network reconstructed light intensity and $S_{snap}$ ($S_{1}\ll S_{snap}$). The reconstructed spectra can be written as follows:(8)$$\hat{f}=f+{\left(S_{snap}-S_{1}\right)}^{-1}n_{s}$$

Based on Equation (8), the SNR of the reconstructed spectrum can be expressed as:(9)$$SNR_{\hat{f}}=10log\left(\frac{\frac{f^{T}f}{n_{s}^{T}{\left(S_{snap}^{T}-S_{1}^{T}\right)}^{-1}{\left(S_{snap}-S_{1}\right)}^{-1}n_{s}}}{\frac{f^{T}f}{n_{s}^{T}n_{s}}}\right)$$

According to our previous works, $SNR_{\hat{f}}$ can be sampled as:(10)$$\begin{array}{c} SNR_{\hat{f}}\geq 10log\left(\frac{{\left(n_{s}^{\prime }\right)}^{T}\left({\left(1-k\right)}^{2}S_{snap}^{T}S_{snap}+\frac{1-k}{k}k^{2}S_{snap}^{T}S_{snap}\right)n_{s}^{\prime }}{{\left(n_{s}^{\prime }\right)}^{T}n_{s}^{\prime }}\right)\\ =10log\left(\frac{{\left(n_{s}^{\prime }\right)}^{T}S_{snap}^{T}S_{snap}n_{s}^{\prime }}{{\left(n_{s}^{\prime }\right)}^{T}n_{s}^{\prime }}\right)+10log\left(1-k\right) \end{array}$$
where $n_{s}^{\prime }={\left(S_{snap}-S_{1}\right)}^{-1}n_{s}$, k denotes perturbations involving reconstruction errors and actual light intensity.

The results show that the reconstructed SNR of the network decreases by about $10log\left(\frac{1}{1-k}\right)$ (dB) compared with the sub-Hadamard matrix which can accurately obtain the intensity distribution. Compared with a traditional slit spectrometer, it still has obvious advantages. In practice, the intensity of overlapped spectra in the single-path snapshot spectrometer is twice the intensity in the original dual-path snapshot spectrometer. Thus, the actual spectrum can be expressed as:(11)$$\hat{f}=2f+{\left(S_{snap}-S_{1}\right)}^{-1}n_{s}$$

After a similar deduction, the final SNR can be expressed as:(12)$$SNR_{\hat{f}}\geq 10log\left(\frac{{\left(n_{s}^{\prime }\right)}^{T}S_{snap}^{T}S_{snap}n_{s}^{\prime }}{{\left(n_{s}^{\prime }\right)}^{T}n_{s}^{\prime }}\right)+10log\left(1-k\right)+10log\left(2\right)$$

Therefore, the proposed method has certain advantages over the dual-path Sub-s HTS.

## 5. Experiments and Simulations

In order to compare the reconstructed performance of different coding matrices, the full-1 matrix and Hadamard-S matrix are involved. Additionally, four frames CASSI [25] using two-step iterative soft threshold optimization (TwIST) are also involved. We present some reconstructed light intensity results that are different from the testing dataset [26,27] and calculate its peak signal-to-noise ratio (PSNR). The reconstructed results are shown in Figure 5. The resolution of reconstructed image is 127 × 127. The comparison of PSNR and reconstruction time of different scenes are shown in Table 2.

It can be seen that the performance of the proposed method is stable under different conditions, and the reconstruction quality of the proposed method is better than that of the four frames CS in the case of a single frame. The testing images are not involved in the training data, and the reconstructed results can also get the expected results, which shows that the network has good generalization ability. Additionally, we can find that the PSNR of reconstructed full-1 coding is better than Hadamard-S coding since the full-1 coding can use the full information and bring higher throughput. However, the full-1 matrix is an irreversible matrix and cannot ensure stable SNR boosting.

We bought a desktop on JD.com with Intel Core i7 8700K 4.3Ghz, manufactured by Intel, Santa Clara, CA, USA; 16GB memory, manufactured by Corsair Memory, Fremont, CA, USA; NVIDIA GeForce GTX 1060 6GB, manufactured by NVIDIA, Santa Clara, CA, USA; The reconstructed time is 0.075 s. It is 20 times faster than traditional methods without additional hardware acceleration. If we use a higher-performance GPU such as RTX TITAN, the reconstructed speed will be significantly improved, which can basically meet the real-time requirements.

In the training strategy, we used the Adam algorithm for gradient optimization; the initial learning rate was 0.001, the whole training process lasted about 150 epochs when reaching convergence and batch size was set to 4. The convolution kernel size of the spectral dimension was modulated according to the dispersion length of the input spectral image.

To verify the effectiveness of reconstructed light intensity, we put the system into implementation and show it in Figure 6. The spectrum with band-pass filters and the whole spectrum were trained and tested in the actual optical system to test the influence of overlapping degrees on the reconstructed effect. In the implementation, the incident light is encoded by a digital micromirror device (DMD). The DMD (ED01N) has a resolution of 1024 × 768 and a pixel size of 13.68 μm × 13.68 μm. The camera (Basler acA1920-155 µm) has a resolution of 1920 × 1200 and a pixel size of 5.86 μm × 5.86 μm, and the dispersion grating is THORLABS GT25-03(300 Grooves/mm, 17.5 deg).

We also tested the light intensity reconstructed performance by increasing spectral bands, and the results are shown in Figure 7. A bandpass filter with 100 nm (400–500 nm) was used to truncate the spectrum, and the results are shown in the first row of Figure 7. The second row of Figure 7 is the result without any filter. The PSNR of reconstructed light intensity with filter is 25.76 dB and 20.26 dB without the filter.

As shown in Figure 7, it can be seen that the proposed method can still reconstruct the light intensity well. However, with the increase of spectral bands, the overlapping becomes more serious, and the reconstructed quality will decrease.

In practice, the Sub-s HTS will lose half throughput since it is the extra imaging path, which will have a negative impact on the SNR of the reconstructed spectrum. According to the theoretical deduction, the SNR decline is 10log(2) (dB). Thus, we compared the slit-based spectrometer, the Sub-s HTS [19] and the proposed net-based HTS through simulation and show the results in Figure 8.

The results show that the proposed net-based HTS can achieve higher SNR results on the premise of reducing an optical path while maintaining snapshot. The SNR of dual-path Sub-s HTS is lower than the proposed net-based HTS which is in accord with the result of theoretical analysis.

The reconstructed quality of light intensity will affect the SNR of the reconstructed spectrum, and with the iteration of network, the quality of reconstructed light intensity will increase. Thus, we compared the SNR of reconstructed spectrum with the iteration of network and Sub-s HTS. The results are shown in Figure 9. It can be seen that as the iterations increase, the SNR of reconstructed spectra increases. As the quality of reconstruction increases, the advantage of light intensity gradually manifests, and the SNR is better than that of Sub-s HTS. This accords with the results of the previous theoretical derivation.

In order to evaluate the performance of the proposed net-based HTS, Sub-s HTS, slit-based spectrometer, we measured the SNR of each method with a low-level light source. In the experiment, we used an absorptive neutral density filter to cover an LED to simulate the low-level light source. We used the slit-based spectrometer to obtain the standard spectrum with the high light of LED and long exposure time. In order to compare the performance of all methods quantitatively, we set the same exposure time in all methods. The captured overlapped spectral image is shown in Figure 10, and the reconstructed spectral data are shown in Figure 11.

The camera (Basler acA1920-155 µm) provides a maximum analog gain setting of 36 dB that increases the brightness of the images output. However, with the increase of analog gain, a lot of electronic noise and temporal dark noise will be introduced. It will be a great influence on the final image quality.

In Figure 11, it can be seen that due to the advantage of light throughput, the net-based HTS has a better SNR than the Sub-s HTS in the case of low-level light environment and strong noise interference. This proves that our system has stronger anti-noise capability. In the case of limited detector capability, our system can greatly improve the SNR of the detected spectrum.

## 6. Conclusions

In this paper, we propose a light intensity recover network (LIRNet) to obtain the original image from overlapped spectra and construct a network-based single-path snapshot Hadamard transform spectrometer (net-based HTS) system. Its advantages over our previous work (Sub-s HTS) lie in the following three aspects. First, we proposed a neural network LIRNet to recover the light intensity distribution from the overlapped dispersive spectra, where the calculation speed is 20 times faster than the traditional methods. Second, the proposed net-based HTS can solve the drawbacks of Sub-s HTS such as the pixel-level registration and the loss of light throughput. Third, thanks to the simple structure of the single-path optical system and the advantage of light throughput, the net-based HTS can obtain a better performance than Sub-s HTS, which further improves its practicability. Additionally, our system can greatly improve the SNR of detected spectrum with limited detector performance.

At the same time, the proposed idea of spectral unmixing can be further developed in the field of computational spectral imaging. We believe that the proposed framework can promote the development and applications of spectral imaging, with reduced hardware and software complexity. In our next research, we will introduce it into hyperspectral imaging to obtain clearer hyperspectral images, which will be an interesting avenue for future work.

## Figures and Tables

**Figure 1 sensors-20-07038-f001:**
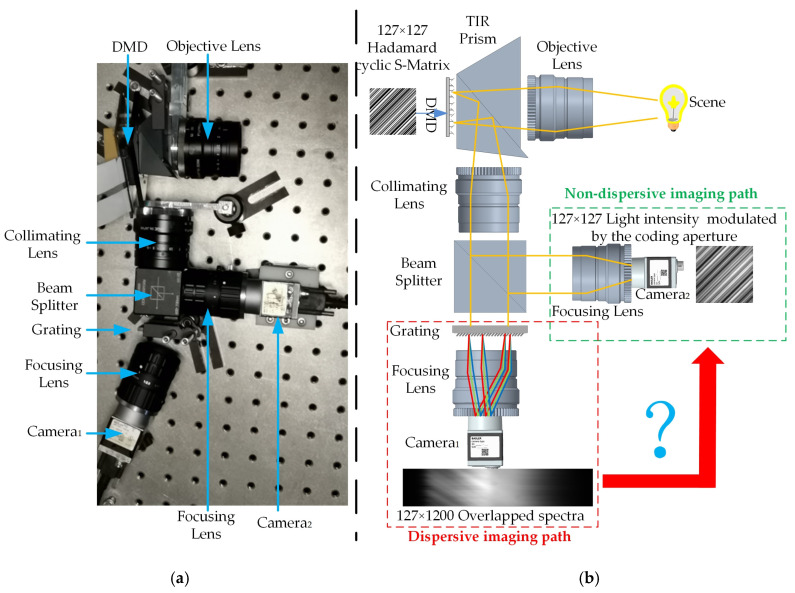
The flowchart of dual-path snapshot Hadamard transform spectrometer (HTS). (**a**) Actual system diagram of HTS. (**b**) Principle system diagram of HTS.

**Figure 2 sensors-20-07038-f002:**
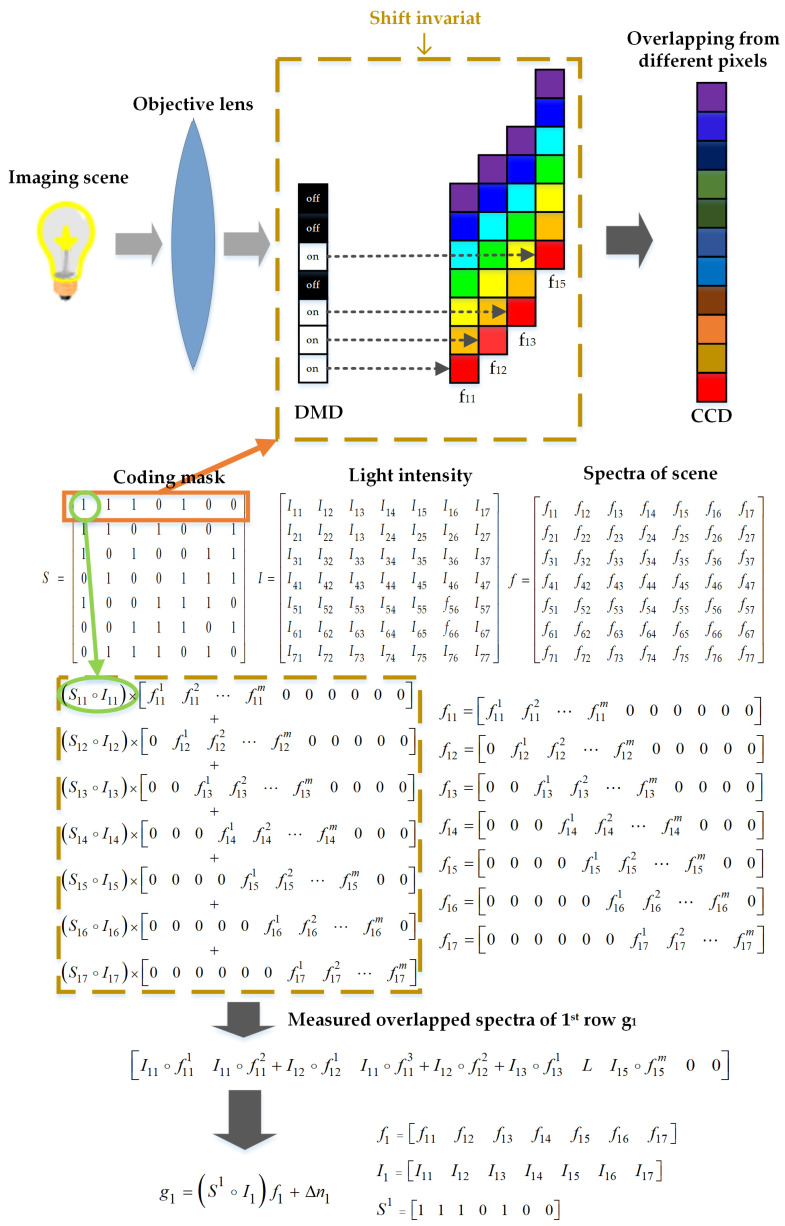
The coding processing of snapshot Hadamard transform spectrometer.

**Figure 3 sensors-20-07038-f003:**
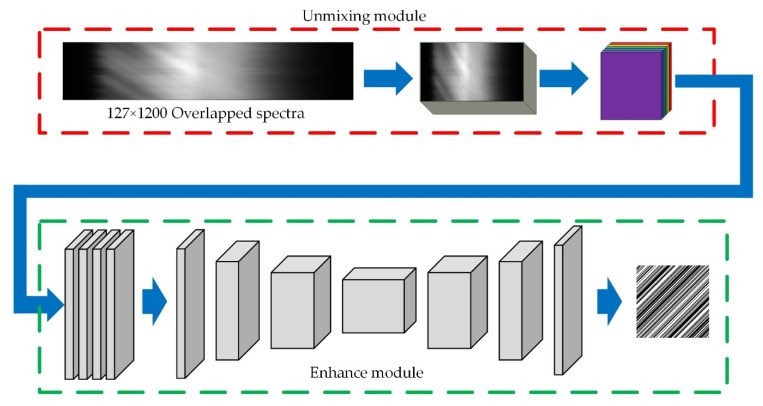
Structural sketch of the network model.

**Figure 4 sensors-20-07038-f004:**
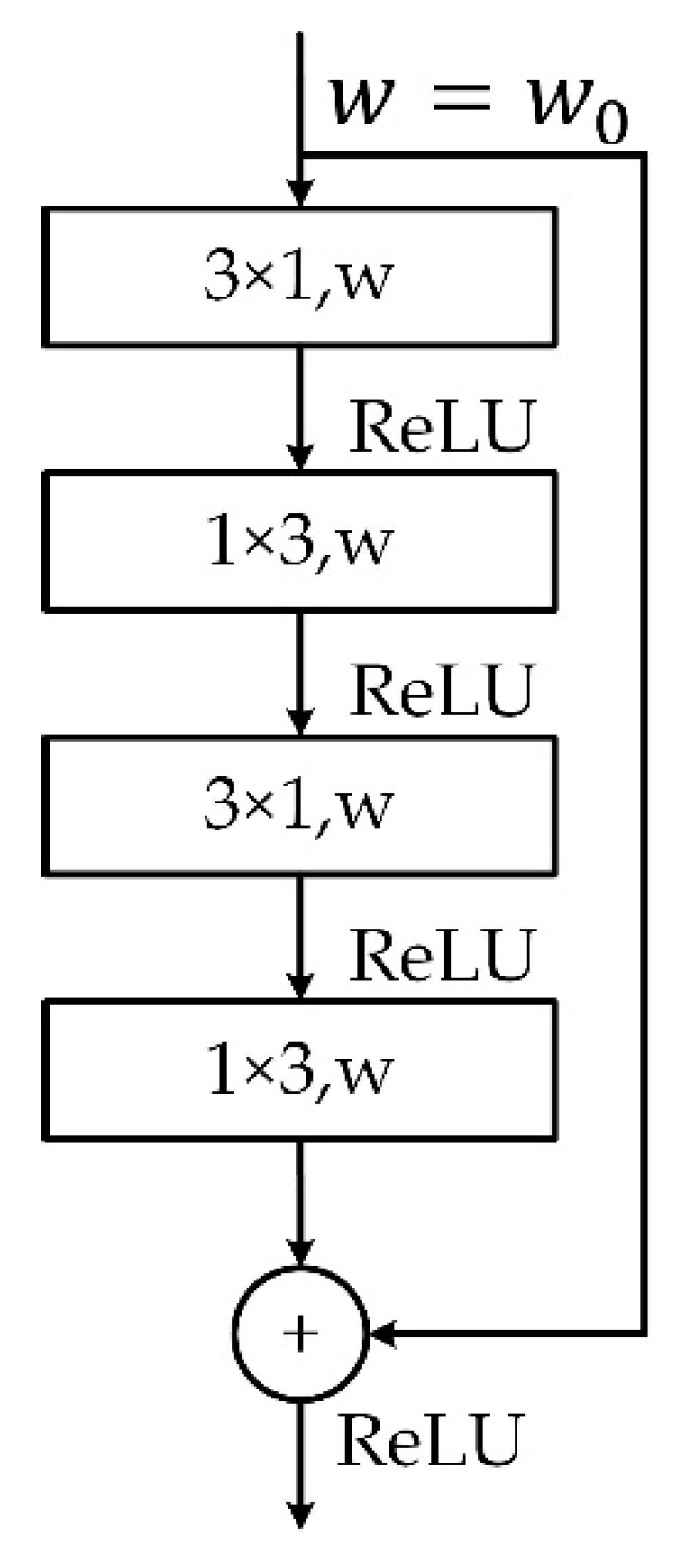
Non-bottleneck 1D structure.

**Figure 5 sensors-20-07038-f005:**
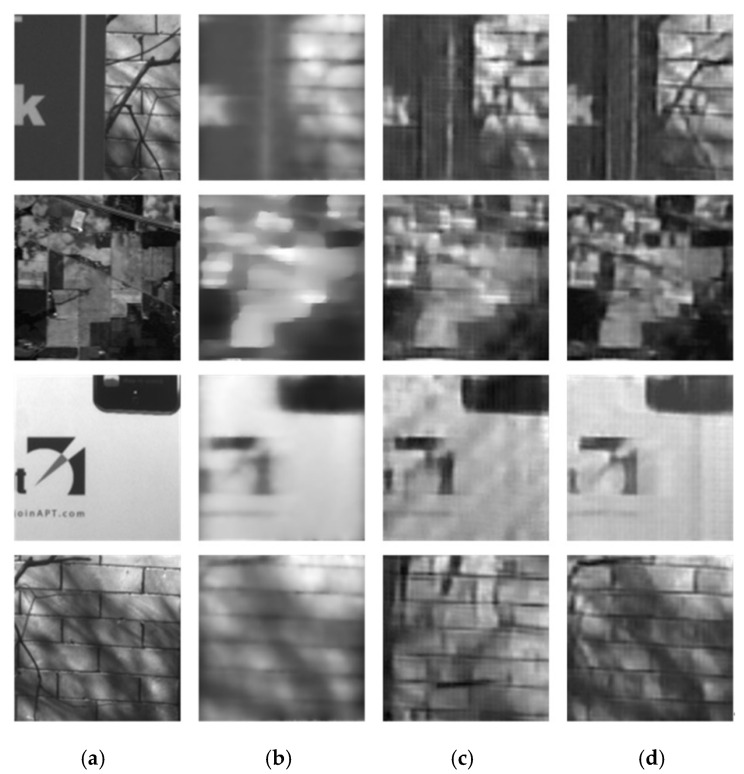
Reconstructed results of different methods. (The resolution of reconstructed image is 127 × 127) (**a**) Ground truth. (**b**) Reconstructed results of coded aperture snapshot spectral imager (CASSI) 4 Frame. (**c**) Reconstructed results of the full-1 matrix (our method). (**d**) results of Hadamard-S matrix (our method).

**Figure 6 sensors-20-07038-f006:**
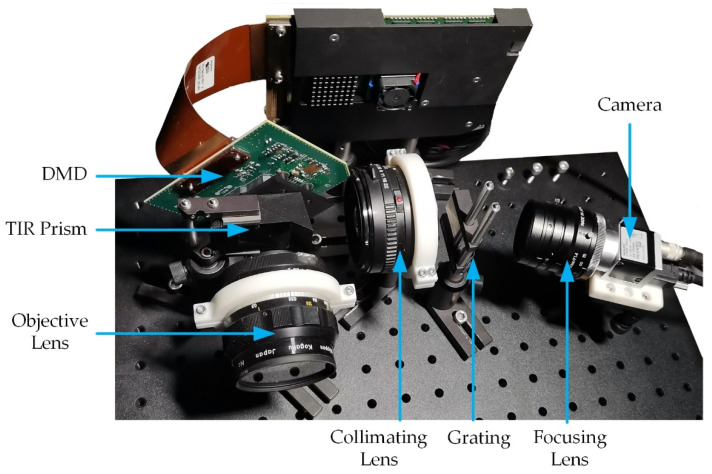
Implementation of single-path snapshot Hadamard transform spectrometer.

**Figure 7 sensors-20-07038-f007:**
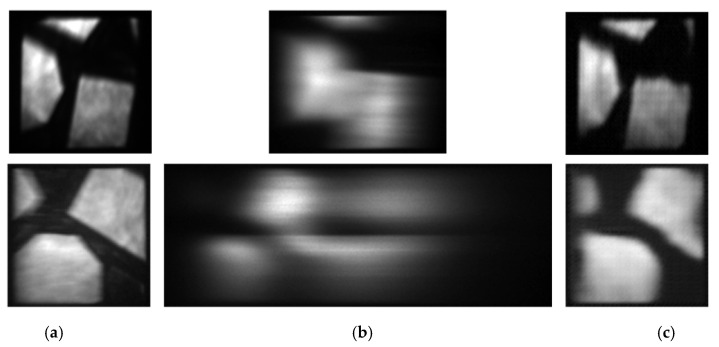
Reconstructed results with different spectral bands. (**a**) Captured light intensity. (**b**) Measured overlapped dispersive spectra. (**c**) Reconstructed light intensity.

**Figure 8 sensors-20-07038-f008:**
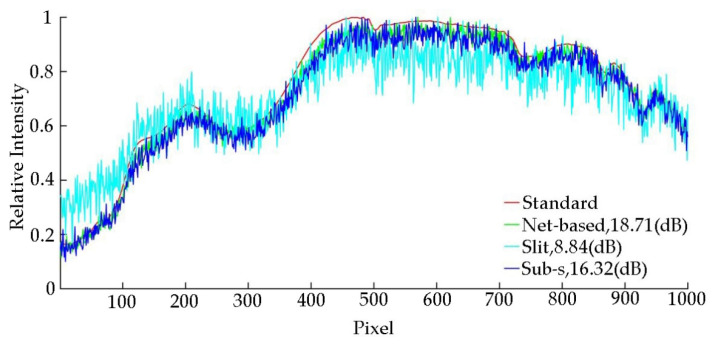
Comparison of different methods.

**Figure 9 sensors-20-07038-f009:**
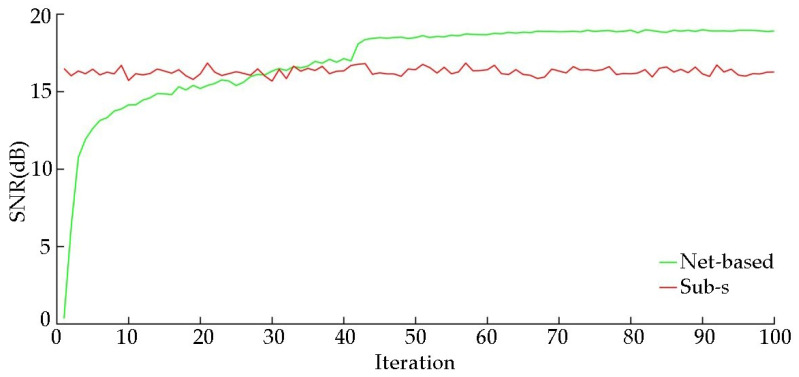
The SNR of the reconstructed spectrum with the iterations number of the network.

**Figure 10 sensors-20-07038-f010:**
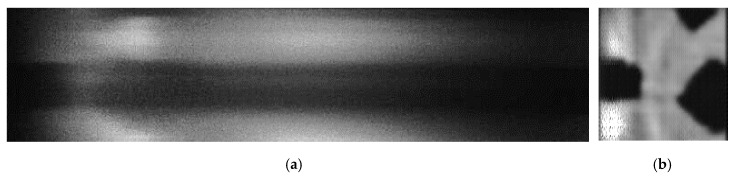
The results of actual experiments with 30 dB (analog gain of camera). (**a**) Overlapped dispersive spectra. (**b**) Reconstructed light intensity.

**Figure 11 sensors-20-07038-f011:**
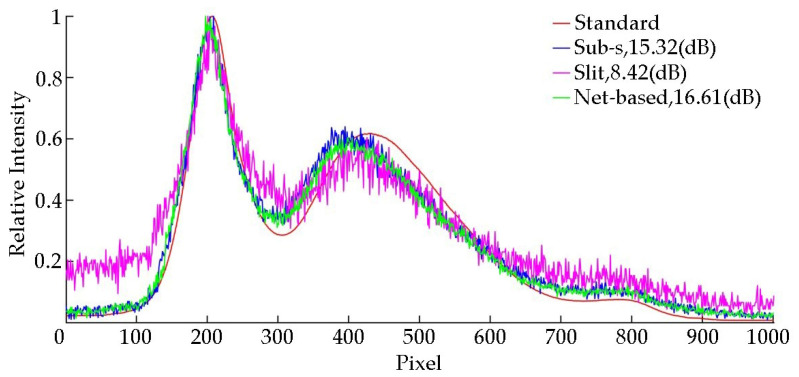
Experimental results of the proposed method, snapshot HTS and slit-based spectrometer with 100 us (exposure time of camera) and 30 dB (analog gain of camera).

**Table 1 sensors-20-07038-t001:** Network structure.

	Layer	Type	Out-F	Out-Res
SIMULATION	1	conv(n × 1)	1	128 × 128
	2	deconv(1 × 1 × n)	n	128 × 128
	3	conv+ReLU	n/2	128 × 128
	4	conv+ReLU	n/4	128 × 128
	5	conv+ReLU	n/8	128 × 128
	6	conv+ReLU	1	128 × 128
ENCODER	7	Downsampler	16	64 × 64
	8	Downsampler	64	32 × 32
	9–13	5 × Non-bt-1D	64	32 × 32
	14	Downsampler	128	16 × 16
	15–22	8 × Non-bt-1D	128	16 × 16
DECODER	23	deconv	64	32 × 32
	24–25	2 × Non-bt-1D	64	32 × 32
	26	deconv	16	64 × 64
	27–28	2 × Non-bt-1D	16	64 × 64
	29	deconv	C	128 × 128

**Table 2 sensors-20-07038-t002:** Comparison of peak signal-to-noise ratio (PSNR) and reconstruction time of different scenes.

Image	CASSI 4 Frame	Full-1 (Ours)	Hadamard-S (Ours)
a	16.09 dB	22.37 dB	21.39 dB
b	12.39 dB	21.48 dB	21.13 dB
c	20.51 dB	20.82 dB	20.60 dB
d	19.44 dB	23.03 dB	22.90 dB
Reconstruction time	1.599 s	0.077 s	0.075 s

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
