# Peer review of "High Sensitivity Snapshot Spectrometer Based on Deep Network Unmixing"

_sensors, 2020, doi:10.3390/s20247038_

Round 1
Reviewer 1 Report
In this manuscript, the authors use a two-module, convolution neural network based approach to carry out the un-mixing of intensity images from the spectral imaging acquired by one of the spectrometer’s optical paths/arms. By doing this, there is no need to use a dedicated arm in the setup to detect intensity, increasing SNR and easing up the two path/two images co-registration problem, among other things (e.g. cost, bulk, etc.). In doing this they describe a valid and novel idea to improve coded aperture, and in particular Hadamard transform, spectroscopy.
There are various typos: just as an example, line 189 “piratical” instead of “practical”, line 250 “can achieves”, etc. Clarity can be improved by avoiding unexplained jargon like “relu” (shall be reLU?), or detailing that 30dB (line 268-9) is the gain of the CCD directly in the caption of figure 11. There are also some expressions that sound weird, like the use (line 20) of “abundant” to describe previous work, and some omissions:
- Line 108: reference to missing red “arrow” in figure 1(b)
- line 114: which equation? Reference is missing
In figure 9, it is shown that with net-based approach, higher number of iterations correspond to higher SNR. Why there is however a sharp increase between iteration 40 and 50 in the green line?
Reviewer 2 Report
The authors present a convolutional neural network-based method for recovering the light intensity distribution from the overlapped dispersive spectra. The presented work follows-up to the authors' previous work – a dual-path snapshot Hadamard Transform Spectrometer (HTS) – and offers the new approach to processing of data.
The topic is within the scope of the journal and should be interesting for readers. The paper is well prepared and structured. The text is entirely understandable. I have only one minor comment.
Minor Comments:
- In Table 2 (page 8, line 207), there is shown a comparison of PSNR of the proposed method and two other state-of-the-art methods. Could You also present a comparison of computational demands (for example the time consumption of image processing for selected pictures) of the methods?
Reviewer 3 Report
The article present interesting new approach in using deep learning in the new light sensor design or spectrometer construction and from scientific point of view the proposed approach looks sound. However, the paper requires major revision before it can be published. A number of problems with the article should be resolved:
- It should be the last comment usually, but the extensive language editing is necessary. I'm sorry to mention it first, but the article is hard to read and some sentences meaning is lost for me. Meaning of the "snapshot" in the title, article text and abstract is not clear in this context. The overall language and style is good, but there is a lot of easy-to-catch language mistakes in the text and some words describing technical aspects are chosen not so good in translation or writing probably. The meaning of some words in the context is also lost for me.
- Quality of figure should be also improved, there is a lot of image compression artifacts on it, the text font is too small everywhere. The caption of figures should include description of all subfigures and be in general more informative, it should contain enough information to be useful for the reader standalone, without the text of article. I would also suggest using of the same font with the similar size for all images in text, each one uses its own font parameters now.
- Figure 6 is summarizing all the comment above: it has a lot of compression artefacts, its too small to be clearly seen in detail, there is no text at all describing what is shown, and it has too short image caption with the language mistake. Please do not do things this way.
- The conclusions are too short, it is hard to tell what this article about from the conclusions, not mentioning the language mistakes. The major results, and what is quantitive difference between the proposed design and the state-of-the art solutions will be welcome here.
- The major problems of state-of-the art solutions that motivated the development should be mentioned somewhere in more clear way.
- The performance of the developed system should be described more clear, it looks that Figure 11 has the very short describing it part in the text and there is no discussion, whether the obtained results are good, bad, or irrelevant. The results part looks abandoned mid-word and incomplete now.
- The major part of the articles in the reference section looks a little bit old for me. The state-of-the art in this area of research can be not so well covered there.
Round 2
Reviewer 3 Report
After the revision the quality of article was improved. It still has some minor issues with english language and too short image captions that can be improved, but in general it is ready to be published. So article can be accepted in the present form.